# Can Accessing the Internet while Travelling Encourage Commuters to Use Public Transport Regardless of Their Attitude?

**Jinhyun Hong \*, David Philip McArthur and Mark Livingston**

Department of Urban Studies, The University of Glasgow, Glasgow G12 8QQ, UK;
David.Mcarthur@glasgow.ac.uk (D.P.M.); Mark.Livingston@glasgow.ac.uk (M.L.)
\* Correspondence: Jinhyun.Hong@glasgow.ac.uk; Tel.: +44-(0)141-330-7652

**Abstract:** Due to advances in technology (in particular the Internet), people have become less restricted by space and time, and can use travel time more productively by using their Internet-connected mobile devices on the move. Some operators provided Internet access on public transport to increase ridership. This has been shown to increase ridership, however it is not clear if it can induce people who prefer private cars to public transport to consider using public transport. In this paper, we examine the relationship between the frequency of using the Internet while commuting or travelling, and commuting mode choice, and how this relationship varies for people who have different attitudes toward public transport. Our results show that commuters who use the Internet frequently on the move tend to use public transport more. In addition, this association is significant for those who prefer private cars to public transport, showing the potential effectiveness of new technology in generating new riders.

**Keywords:** Internet use; commuting behaviour; attitude

## 1. Introduction

The relationship between information and communication technologies (ICT) and travel has been examined for several decades because of the impact these technologies have on the way people live, work and travel [1–8]. Interestingly, some empirical evidence implies that ubiquitous access to the Internet through different mobile devices has the potential to increase public transport ridership by changing people's perceptions about travel time. Traditionally, travel time has been considered as a cost (wasted time) in economic appraisals and evaluations [9], often making public transport less attractive than private cars. Nowadays, people carry out multiple tasks while traveling, many of which require Internet access, changing travel time from wasted time to more productive time [10,11]. Public transport users may obtain more benefits from this new technology than drivers because they are less restricted both mentally and physically, even though some drivers enjoy the Internet (e.g., getting real time traffic information, listening to the radio, etc.) while driving.

Based on the above evidence, policies such as installing Wi-Fi on trains and buses to increase ridership have been proposed (e.g., Amtrack in California, USA, provided free Wi-Fi services on trains on the California Capitol Route in 2011 [12]). This has helped to generate an expectation of less car dependency in the future. To properly evaluate the potential of this approach, it is important to understand if it can encourage people who do not particularly like public transport compared to private cars to use it more. It may be the case that ubiquitous Internet access while travelling will increase the attractiveness of public transport for those who already have a positive disposition towards public transport with little or no effect on those who do not. If this is true, then the potential of providing

Internet access to travellers would have only a limited ability to affect mode change away from driving. Unfortunately, empirical studies are scarce, partly due to data limitations. In this paper, we aim to address the above issue by examining how the relationship between Internet use while commuting or travelling and commuting mode choice varies according to attitude towards public transport. It should be noted that we do not particularly examine if commuters use WiFi on buses or trains. Rather, we investigate how the frequency of Internet use on the move is related to the commuting mode choice across travellers who have different attitudes towards public transport. Although we do not consider the provision of WiFi on public transport vehicles directly, we believe our results will provide valid evidence on its usefulness. Specifically, two research questions are investigated: 1) how the frequency of Internet use while commuting or travelling on buses, cars or trains is associated with commuting mode choice (i.e., car versus public transport); and 2) how its relationship changes between commuters who have different attitudes towards public transport (i.e., those who are favourable to public transport compared to private cars versus others).

## 2. Literature Review

The rapid spread of Internet-connected mobile devices has led to significant changes in people's lives and travel behaviour. One consequence has been people becoming less restricted by space and time. For example, people work at home, in the public realm, in cafes and even on the move. Moreover, the use of travel time and its value has been changing over time in the digital age [13,14]. This new phenomenon has the potential to change the relative attractiveness of different transport modes, thereby influencing travellers' mode choices.

Specifically, having mobile Internet access could change or enhance the perception of travel time from a cost to some sort of benefit. For example, Jain and Lyons [15] argued that travel time itself could provide benefits to some people e.g., giving time to prepare for meetings, change mood, etc. Ben-Elia, et al. [16] mentioned that the level of tolerance for travel time could be higher than in the past due to the effects of ICT, making travel time more useful and productive. Lyons, Jain, Susilo and Atkins [13] found that more rail passengers use ICT while traveling in 2010 than in 2004, with a 60% increase in the number of people bringing and using a laptop. Moreover, the proportion of passengers who made worthwhile use of their travel time on the train increased between 2004 and 2010 while the number of passengers who said that their travel time was wasted time decreased over six years. Banerjee and Kanafani [17] examined the potential effects of working on the train on the opportunity cost of travel time, commonly referred to as the value of travel time or VOTT. Their numerical example showed that as the efficiency of working on the train increases, the VOTT decreases while the utility increases. This implies that a wireless Internet connection on public transport could change its relative attractiveness by lowering the value of travel time. However, they also indicated that the effects of Internet services will vary across riders depending on their travel and facility conditions (e.g., short commutes, crowded buses or trains, etc.).

Unsurprisingly, this impact can vary across transport modes. For example, drivers are more restricted in using the Internet or mobile devices while traveling compared to public transport users. Therefore, the benefits of Internet use on the VOTT may differ for public transport and private car users, potentially resulting in increased public transport ridership. Several empirical studies have supported the above hypotheses. Dong, Mokhtarian, Circella and Allison [12] examined how Wi-Fi influences public transport ridership and how its impacts vary between current and new riders. They found that free Wi-Fi on the train increases train trip frequencies for both new and continuing riders. Zhang, et al. [18] also examined how wireless Internet service (WIS) influences travel and work performance of business travellers in the Netherlands. Their results supported a positive effect of WIS, implying increased utility of train use due to the WIS. Clayton, et al. [19] also found that several travel-time activities (e.g., window gazing, listening to music and checking emails) on the bus are positively associated with journey experience although they are not the primary factors. In sum, these results imply that Internet use could provide higher benefits to public transport users than drivers,

encouraging travellers to use public transport rather than private cars. However, it is also possible that Internet use with mobile devices provides some benefits for drivers. For example, a car can be a moving office and provide flexibility to mobile professionals [20,21]. In addition, people can potentially travel more efficiently or change travel modes by accessing real-time travel information [22,23]. Hong and Thakuriah [7] indicated that a substantial number of workers in their sample obtain travel information through apps, websites and systems when they make car trips. Combined with the fact that travel attitudes are very important determinants of travel patterns [24–26], it implies that people who mostly drive or who prefer private cars to public transport may continue using their cars for their daily trips due to the benefits from this new technology. If this effect is true and strong enough, only a limited number of drivers may change their travel mode, resulting in limited new public transport riders. Therefore, it is important for planners and policy makers to understand how this technology could influence travel mode choice of people who do not like public transport compared to private cars.

In this study, we utilised commuters' attitude towards public transport and driving to create two different groups i.e., those who have affirmative attitude towards public transport compared to private cars and those who do not. Then, we examine how the relationship between the frequency of Internet use while commuting or travelling and commuting mode choice varies between these two groups. Moreover, we checked for potential endogeneity to obtain more robust results. Some scholars argued that ICT use and travel behaviour could have bidirectional associations [7,27]. For example, Miranda-Moreno, et al. [28] argued that unmeasured factors could affect both ICT use and travel outcomes, resulting in an incorrect relationship between them unless this endogeneity impact is controlled for. Their analytical results support the assumption, implying the importance of using appropriate analytical models. We assumed that Internet use influences mode choice, and utilised a recursive bivariate probit (RBVP) model to test for the presence of endogeneity between the frequency of Internet use on the move and commuting mode choice. The details of the model can be found in the Monfardini and Radice [29] study.

## 3. Materials and Methods

### 3.1. Data

The Urban Big Data Centre (UBDC) at the University of Glasgow conducted a household survey of a representative group of residents of the Glasgow and Clyde Valley Planning area. The survey area consists of eight local authorities (East Dunbartonshire, East Renfrewshire, Glasgow City, Inverclyde, North Lanarkshire, Renfrewshire, South Lanarkshire and West Dunbartonshire) and has a third of Scotland's total population. This area is commonly considered to be the definition of Greater Glasgow. The survey was conducted through face-to-face interviews over an eight-month period in 2015, and includes a total of 2095 people from 1511 households. Data is available for researchers upon request (ubdc.ac.uk).

The survey includes several ICT-related questions. One question asks "Do you use the Internet at all these days, either for your work or for your personal use?". If the respondent answers either "personal use" or "both work and personal use", the subsequent question asks "How often do you use the Internet while you are commuting or travelling on buses, cars or trains?". The response is recorded by a four-point Likert scale, anchored by "never" and "almost always". We combined answers from these two questions and created an *Internet use while travelling* variable. We recategorized responses from the variable into two categories (i.e., 0: "never + rarely" and 1: "sometimes + almost always") to investigate a potential endogeneity issue. A more detailed explanation about endogeneity is given in Section 3.2.

The survey also includes several attitudinal questions about specific transport modes (i.e., walking, cycling, public transport and car). For our research, we used two of the attitudinal questions about public transport and private cars. Specifically, the survey asks how much the respondent agrees or disagrees with the statement "For me, to use public transport for regular or daily journeys is something

I like". The same question is asked for driving a car ("For me, to drive a car for regular or daily journeys is something I like"). The answers are measured on a five-point ordinal scale, from "strongly disagree" (coded 1) to "strongly agree" (coded 5). We calculated the difference between these two responses (Att_PT—Att_Drive, see Table 1) and created a *Pro-public transport* variable. If participants have a negative value of *Pro-public transport*, we referred to them as "Dis-likers". On the other hand, we referred to people who have a positive value (including 0) of *Pro-public transport* as "Likers".

**Table 1.** Descriptive statistics.

| | Full Sample | | Likers | | Dis-Likers | |
|---|---|---|---|---|---|---|
| | **Mean** | **SD** | **Mean** | **SD** | **Mean** | **SD** |
| **Socio-demographics** | | | | | | |
| Age | 40.72 | 12.91 | 39.82 | 13.82 | 41.29 | 12.27 |
| Gender (Male = 1) | 47% | | 43% | | 50% | |
| Education (Higher education = 1) | 40% | | 36% | | 43% | |
| Household size | 2.82 | 1.29 | 2.61 | 1.24 | 2.97 | 1.31 |
| Work status (Work = 1) | 92% | | 90% | | 94% | |
| Drive licence (Own = 1) | 79% | | 57% | | 94% | |
| *Residential locations* | | | | | | |
| Large urban areas | 57% | | 61% | | 55% | |
| Other urban areas | 30% | | 28% | | 31% | |
| Towns and rural areas | 13% | | 11% | | 14% | |
| **Attitude** | | | | | | |
| *Att_PT—"To use public transport for regular or daily journey is something I like"* | | | | | | |
| Strongly disagree (1) | 19% | | | | | |
| Disagree (2) | 22% | | | | | |
| Neutral (3) | 21% | | | | | |
| Agree (4) | 29% | | | | | |
| Strongly agree (5) | 9% | | | | | |
| *Att_Drive—"To drive a car for regular or daily journeys is something I like"* | | | | | | |
| Strongly disagree (1) | 13% | | | | | |
| Disagree (2) | 7% | | | | | |
| Neutral (3) | 11% | | | | | |
| Agree (4) | 30% | | | | | |
| Strongly agree (5) | 39% | | | | | |
| Pro-public transport (Att_PT—Att_Drive) | −0.88 | 2.14 | | | | |
| **Internet use while travelling** | | | | | | |
| Never and rarely | 57% | | 50% | | 61% | |
| Sometimes and almost always | 43% | | 50% | | 39% | |
| Sample size | 764 | | 300 | | 464 | |

The survey includes several questions about commuting behaviour. One question asks "How do you usually travel to work (or school/college, university if in full time education)?" For this study, we extracted commuters and full-time students who use either cars (private cars, motorcycle and taxi) or public transport (bus, rail and subway). This is our dependent variable, indicating whether commuters use a car (reference group) or public transport. In addition, we included several socio-demographic factors (age, gender, education level, work status, possession of a driving license) and variables describing respondents' residential locations (large urban areas, other urban areas and town or rural areas) as these factors are known to influence commuting behaviour based on previous studies.

### 3.2. Analytical Models

We used binary logistic regression models to investigate the relationship between the frequency of Internet use while travelling and commuting mode choice (car vs. public transport). To examine

how their relationship varies according to attitude towards public transport compared to private cars, we created two groups (i.e., Likers: people who prefer public transport to private cars; and Dis-Likers: people who do not prefer public transport to private cars) and ran a separate model for each group. In sum, we ran the same model for the full sample, the Likers and the Dis-Likers.

Before using a simple binary logistic regression model, we checked for the potential methodological issue of endogeneity in the ICT-travel behaviour analysis. Namely, people who use ICT more could travel more due to the easy access to new information. On the other hand, people who travel often may use ICT more to find real time traffic information or available public transport services. If this is true and its impact is not well treated in the analytical model, the estimates from the model will be biased. In our case, we hypothesize that Internet use while travelling could increase the relative utility of public transport to private cars, leading to increased public transport ridership. However, it could be the case that people who use public transport could be more likely to use the Internet on the move compared to drivers for different purposes (e.g., doing work, watching video clips, reading news, etc.) because they are mentally and physically less restricted. Therefore, we need to check if an endogeneity issue exists when examining the relationship between Internet use while travelling and commuting mode choice. For this purpose, we used a recursive bivariate probit (RBVP) model with an instrumental variable. The model assumes that two binary logistic regression models are correlated as shown below:

$$
\begin{aligned}
y_i^* &= \alpha + \beta_{SD}^\top X_{SDi} + \beta_{Att} X_{Atti} + \beta_{Internet} D_i + \varepsilon_{1i} \\
D_i^* &= \gamma + \gamma_{SD}^\top X_{SDi} + \gamma_{Att} X_{Atti} + \gamma_{SK} X_{SKi} + \varepsilon_{2i}, \text{ for } i = 1, \dots n \text{ (\# of observations)} \\
\begin{pmatrix} \varepsilon_1 \\ \varepsilon_2 \end{pmatrix} &\sim \begin{pmatrix} 1 & \rho \\ \rho & 1 \end{pmatrix}
\end{aligned}
\tag{1}
$$

where, $y^*$ *and* $D^*$ are latent variables for commuting mode choice and Internet use while travelling, and $X_{SD}$, $X_{Att}$ *and* $X_{SK}$ represent socio-demographic factors (including residential locations), attitudes, and Internet skill (instrumental variable), respectively.

To identify the model, we need an instrumental variable which is correlated with *Internet use while travelling* ($D^*$) but not commuting mode choice ($y^*$). The iMCD survey includes ICT-literacy questions and we used one of them as an instrumental variable. The survey asks how confident the respondent is at fixing computer-related problems such as network issues or getting a new device to work. The answer is measured by a four-point Likert scale, anchored by "not at all confident" and "very confident". We checked that this variable was statistically significantly associated with *Internet use while travelling* but not with commuting mode choice. If $\rho = 0$, there is no endogeneity issue. The *biprobit* command in Stata provides a likelihood ratio test result for a RBVP model by comparing the log likelihood of the RBVP model with the sum of log likelihoods from two separate univariate probit models. Our analyses showed that there is no endogeneity impact for all three cases (i.e., full sample (Chi-squre: 0.49, p-value: 0.48), Likers (Chi-square: 0.00, p-value: 0.98) and Dis-Likers (Chi-square: 0.73, p-value: 0.39)). Therefore, we decided to use a simple logistic model for our analyses. Our final model can be written as below:

$$
\begin{aligned}
\Pr(y_i = 1 | X_{SDi}, X_{Atti}, D_i) &= logit^{-1}\big(\alpha + \beta_{SD}^\top X_{SDi} + \beta_{Att} X_{Atti} + \beta_{Internet} D_i\big), \\
&\quad for \ i = 1, \dots n
\end{aligned}
\tag{2}
$$

## 4. Results

The descriptive statistics for our full sample, Likers and Dis-Likers are shown in Table 1. In general, they have similar socio-demographic characteristics. The average age of our respondents is about 41 years. In total, 47% of them are male and 40% have a higher education degree. On average, there are 2.82 members per household and 92% of them are workers. Note that we are only considering commuters and full-time students for our analyses. In total, 79% of our observations hold a valid driving licence with a higher percentage for Dis-Likers than Likers. About 60%, 30% and 10% of the

respondents live in large urban areas, other urban areas, and towns and rural areas, respectively. The statistics also show that more people who prefer public transport to private cars live in large urban areas than those who do not. This implies that attitudes towards public transport might be associated with residential location choices, supporting the argument for a self-selection impact in the travel-land use analyses.

In total, 41% of our observations do not like public transport for their regular or daily journeys (participants who answered "Strongly disagree" or "Disagree") while only 20% do not like driving. About 60% of observations never or rarely use the Internet while commuting or travelling on buses, cars or trains. Interestingly, a higher percentage of Likers (50%) use the Internet while travelling than Dis-Likers (39%).

Figure 1 provides a clearer picture about the relationship between Internet use while travelling and commuting mode choice between two different groups. It shows that there is no clear pattern among Likers while commuters who use the Internet on the move are more likely to use public transport among Dis-Likers. The proportion of Likers using public transport is already high, showing the importance of attitudes on the commuting mode choice. This may be the reason why there is no clear difference between people who use the Internet frequently on the move and those who do not among Likers. This implies that providing Internet access on public transport could encourage people who do not like public transport compared to private cars to use it.

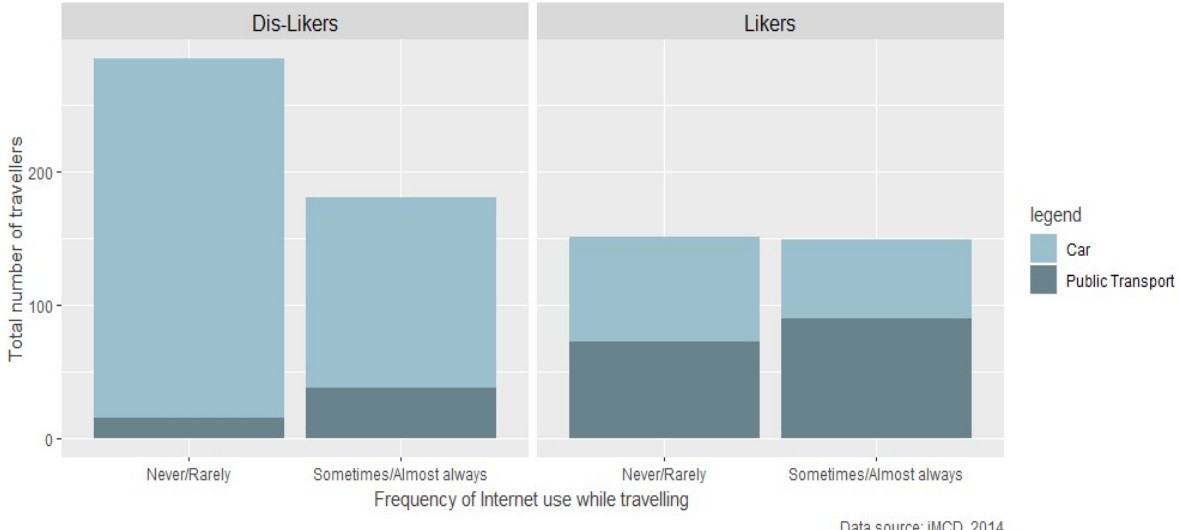

**Figure 1.** Frequency of Internet use while travelling and commuting mode choice.

To further examine their relationship, we ran binary logistic regression models. First, we ran models with the full-sample and the results are shown in Table 2. Most socio-demographic factors show consistent results compared to previous empirical studies. As people age, they tend to drive more rather than take public transport for their commuting. This could be due to their physical condition. However, this association is only significant when considering both socio-demographic factors and attitudes (Model 2). Similarly, highly educated people are more likely to use public transport but its significant association with commuting mode choice disappears when considering the frequency of Internet use while travelling (Model 3). Compared to students, workers are more likely to drive. Driving a car is, in general, more expensive and workers are more affluent than students. In addition, commuters who hold a valid driving licence tend to drive more than riding public transport. Models 3 shows that people living in other urban areas compared to those residing in large urban areas are less likely to use public transport while controlling for socio-demographic factors, attitudes and the frequency of Internet use while travelling. Large urban areas have a better public transport system than other urban areas, enabling people to use public transport.

**Table 2.** Results of the relationship between Internet use while travelling and commuting mode choice for full samples (reference = car).

| | Model 1 SD | | | Model 2 SD+Att | | | Model 3 SD+Att+Int | | |
|---|---|---|---|---|---|---|---|---|---|
| | β | SE | | β | SE | | β | SE | |
| Intercept | 3.33 | 0.52 | *** | 2.88 | 0.55 | *** | 2.25 | 0.58 | *** |
| **Socio-demographics (SD)** | | | | | | | | | |
| Age | −0.01 | 0.01 | . | −0.03 | 0.01 | ** | −0.02 | 0.01 | . |
| Gender (Male = 1) | 0.34 | 0.20 | . | 0.33 | 0.22 | | 0.32 | 0.22 | |
| Education (Higher education = 1) | 0.49 | 0.21 | * | 0.50 | 0.23 | * | 0.39 | 0.23 | . |
| Household size | −0.25 | 0.08 | ** | −0.14 | 0.09 | | −0.15 | 0.09 | . |
| Work status (Work = 1) | −1.19 | 0.37 | ** | −1.17 | 0.39 | ** | −1.20 | 0.39 | ** |
| Drive licence (Own = 1) | −2.85 | 0.24 | *** | −1.61 | 0.28 | *** | −1.66 | 0.28 | *** |
| *Residential locations (ref: Large urban areas)* | | | | | | | | | |
| Other urban areas | −0.42 | 0.06 | . | −0.46 | 0.25 | . | −0.54 | 0.25 | * |
| Towns and rural areas | −0.34 | 0.32 | | −0.35 | 0.34 | | −0.47 | 0.35 | |
| **Attitude towards public transport (Att)** | | | | | | | | | |
| Pro-public transport | | | | 0.58 | 0.07 | *** | 0.57 | 0.07 | *** |
| **Internet use while travelling (Int)** (ref: never and rarely) | | | | | | | | | |
| Sometimes and almost always | | | | | | | 0.78 | 0.22 | *** |
| McFadden's R-squared | 0.26 | | | 0.37 | | | 0.38 | | |
| Sample size | 764 | | | | | | | | |

. Significant at the 0.10 level of significance; * Significant at the 0.05 level of significance; ** Significant at the 0.01 level of significance; *** Significant at the 0.001 level of significance

The result shows a positive association between *Pro-public transport* and public transport use, and its association is significant at the 0.001 level of significance. That is, people who prefer public transport to private cars are more likely to use public transport, and this result is consistent with previous studies. This also confirms that attitude is a very important determinant of commuting mode choice. Finally, model 3 shows that people who use the Internet frequently while travelling are more likely to use public transport compared to private cars, and its association is statistically significant at the 0.001 level of significance. As discussed, Internet use could influence VOTT and its impact might be different between public transport riders and drivers. This implies that new technology could be a useful policy tool for attracting people to use public transport.

To investigate if this association is consistent regardless of commuters' attitude, we ran the models for two groups (i.e., Likers and Dis-Likers), and results are presented in Table 3. Since the sample size is relatively small, the models show relatively fewer significant associations. For both groups, having a valid drive licence is a very important determinant of commuting mode choice. Interestingly, our model for Dis-Liker shows that people who use the Internet sometimes or almost always while travelling are more likely to use public transport. This implies that we may achieve a significant increase in the number of new public transport users through utilising new technology because people who prefer private cars to public transport are highly less likely to use public transport for their commuting. For Likers, Figure 1 also shows that there is no significant difference between people who use the Internet frequently on the move and those who do not, and our model result confirms it. As shown in Table 2, attitude is a very important determinant of commuting mode choice and it may be the case that people who like public transport will use it regardless of their use of the Internet. Figure 2 shows the relative magnitudes of association across different groups. It shows the predicted probabilities of choosing public transport for commuters with different levels of Internet use while travelling and its 95% confidential interval. We can see that its relative effect (i.e., change) becomes greater for Dis-Likers.

**Table 3.** Results of the relationship between Internet use while travelling and commuting mode choice for Likers and Dis-Likers (reference = car).

| | Likers | | | Dis-Likers | | |
|---|---|---|---|---|---|---|
| | β | SE | | β | SE | |
| Intercept | 2.78 | 0.79 | *** | 1.58 | 0.90 | . |
| **Socio-demographics** | | | | | | |
| Age | −0.01 | 0.01 | | −0.02 | 0.02 | |
| Gender (Male = 1) | 0.25 | 0.28 | | 0.64 | 0.35 | . |
| Education (Higher education = 1) | 0.73 | 0.31 | * | −0.04 | 0.35 | |
| Household size | −0.18 | 0.12 | | −0.19 | 0.13 | |
| Work status (Work = 1) | −1.00 | 0.58 | . | −1.32 | 0.55 | * |
| Drive licence (Own = 1) | −2.21 | 0.32 | *** | −2.50 | 0.50 | *** |
| *Residential locations (ref: Large urban areas)* | | | | | | |
| Other urban areas | −0.41 | 0.31 | | −0.62 | 0.41 | |
| Towns and rural areas | −0.61 | 0.45 | | 0.02 | 0.49 | |
| **Internet use while travelling (ref: never and rarely)** | | | | | | |
| Sometimes and almost always | 0.39 | 0.29 | | 1.39 | 0.39 | *** |
| McFadden's R-squared | 0.20 | | | 0.24 | | |
| Sample size | 300 | | | 464 | | |

. Significant at the 0.10 level of significance; * Significant at the 0.05 level of significance; ** Significant at the 0.01 level of significance; *** Significant at the 0.001 level of significance

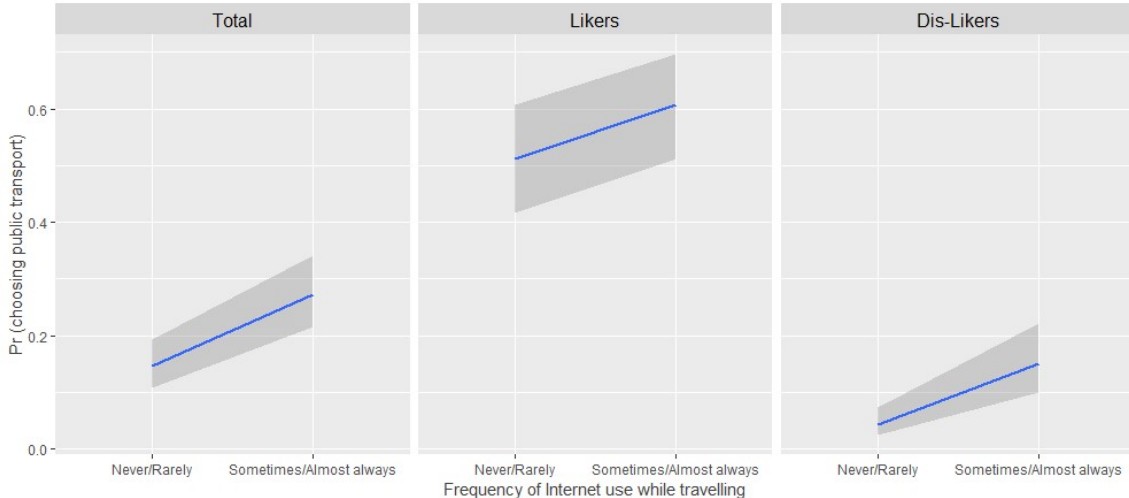

**Figure 2.** The effects of using the Internet while travelling on the probabilities of choosing public transport.

## 5. Conclusions

Access to the Internet has spread rapidly for decades and has changed the way people live and travel. In addition, the increasing use of mobile technologies has changed perceptions about travel time. For example, commuters can work on trains using mobile devices and an Internet connection, making their travel time more productive and reducing the cost of travel time. This could result in modal shift from private cars to public transport. Based on this evidence, some operators made Internet access available on public transport to increase ridership. To evaluate the effectiveness of this new policy, it is critical to understand if Internet use on the move attracts new public transport riders who, in general, do not like public transport compared to private cars. Unfortunately, empirical studies are scarce, limiting our understanding. In this study, we focused on how Internet use while

commuting or travelling is associated with commuting mode choice, and how its relationship varies between commuters with different attitudes towards public transport and private cars.

Our results show that attitude is a very important determinant of commuting mode choice. People who prefer public transport to cars tend to use public transport more than those who do not. This implies the importance of changing commuters' attitude to reduce car dependency. Some empirical studies showed that educating people or making structural changes (e.g., social and individualised marketing campaigns, temporarily providing free public transport tickets) could change peoples' attitudes [30,31], thereby increasing ridership.

Secondly, we found a significant and positive association between the frequency of Internet use while commuting or travelling and using public transport for commuting. This result supports the arguments that the value of travel time has changed due to Internet use on mobile technologies, and its impact is more significant for public transport than cars. This result implies that providing facilities such as free Wi-Fi on trains or buses could increase ridership and reduce car dependency for commuters. In addition, our results also showed that this policy may be effective for commuters who dislike public transport. That is, we did find a significant association between the frequency of Internet use while commuting or travelling and commuting mode choice among people who do not prefer public transport to private cars. Combined with the fact that people who dislike public transport tend to use private cars more often, our results imply that there could be significant increases in public transport ridership and the number of new riders.

Lastly, our results show that residential location is an important determinant of using public transport. Commuters living in large urban areas where the quality of public transport is better are more likely to use public transport than those living in other urban areas.

**Author Contributions:** Conceptualization, J.H. and D.P.M.; Methodology, J.H.; Validation, D.P.M. and M.L.; Data Curation, J.H. and M.L.; Writing—Original Draft Preparation, J.H.; Writing—Review and Editing, D.P.M. and M.L.; Visualization, J.H.

**Acknowledgments:** This work includes use of the iMCD data collection, which was created by the Urban Big Data Centre at the University of Glasgow, supported by the Economic and Social Research Council (grant numbers ES/L011921/1, ES/S007105/1).

**Conflicts of Interest:** The authors declare no conflict of interest.

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
