# Peer review of "Can Accessing the Internet while Travelling Encourage Commuters to Use Public Transport Regardless of Their Attitude?"

_sustainability, doi:10.3390/su11123281_

Reviewer 1 Report

The article deals with an interesting topic and fits for Journal’s aim. The issue discussed in this study is original with reference to contemporary scientific discussion on the implementation of the public transport paradigms. The study highlights remarkable points of view and critically mention issues in the definition and applicability of public transport.

In my opinion, the study should not be accepted for publication in its present version since it presents a number of caveats. In a reviser version of the papier, the author should carefully address the following points.

1.      Introduction should inform about the aim of manuscript.

2.      The literature review should contain references concerning the methodology used in the study. Moreover, I would suggest the author put in evidence analogies and differences of related studies with respect to the methodology which implements in this study.

3.   The results should be clearly present (which municipalities are functional or potential functional area).

4.      I would suggest the author make the reader aware of the possible reasons of the results.

Author Response

We would like to thank you for what was obviously a very careful reading of our paper, and for the comments which you have provided. Please see the below responses for your comments. 

Comment 1: “Introduction should inform about the aim of manuscript.”

Authors: We included the aim of our paper in page 1 (“Unfortunately, empirical studies are scarce, partly due to data limitations. In this paper, we aim to address the above issue by examining how the relationship between Internet use while commuting or travelling and commuting mode choice varies according to attitude towards public transport”)

 Comment 2: “The literature review should contain references concerning the methodology used in the study. Moreover, I would suggest the author put in evidence analogies and differences of related studies with respect to the methodology which implements in this study.”

Authors: We included some references related to the endogeneity effect in the ICT-travel behaviour analyses in Literature review chapter. In addition, we indicated why we used a bivariate probit model with a reference to support its appropriateness (See page 3 “Moreover, we checked for potential endogeneity to obtain more robust results. Some scholars argued that ICT use and travel behaviour could have bidirectional associations [7, 27]. For example, Miranda-Moreno, et al. [28] argued that unmeasured factors could affect both ICT use and travel outcomes, resulting in an incorrect relationship between them unless this endogeneity impact is controlled for. Their analytical results support the assumption, implying the importance of using appropriate analytical models. We assumed that Internet use influences mode choice, and utilised a recursive bivariate probit (RBVP) model to test for the presence of endogeneity between the frequency of Internet use on the move and commuting mode choice. The details of the model can be found in the Monfardini and Radice [29]’ study.”).

Most previous studies in our paper did not consider an endogeneity effect because their analyses are not directly associated with ICT use. For example, Dong, Z. et al (2015) utilised a multiple linear regression model to evaluate the impact of free Wi-Fi on the train ridership. They asked train riders what causes them to ride trains more often. One of the options is “Wi-Fi”. Therefore, it is not easy to examine different studies with respect to the methodology which is implemented in our paper.   

 Comment 3.   The results should be clearly present (which municipalities are functional or potential functional area).

 We have added additional explanation about what councils make up the Glasgow and Clyde Valley Area (East Dunbartonshire, East Renfrewshire, Glasgow City, Inverclyde, North Lanarkshire, Renfrewshire, South Lanarkshire and West Dunbartonshire.) We have also clarified that this planning area is one of the common definitions of Greater Glasgow.

Comment 4. “I would suggest the author make the reader aware of the possible reasons of the results.”

Authors: We added several explanations in this revised manuscript.

Reviewer 2 Report

I found really interesting the idea to explore the attractiveness of public transport thanks to internet access regardless of individual attitudes toward it. I just have few remarks:

1.       Probably shorten the title – you can add more keywords instead. For instance, the title can be something like ”Can access to the Internet on public transport encourage more commuters to use it regardless of their attitude?” or something similar. Think about it.

2.       The keyword ”internet use while commuting” could be replaced with something else – it is more a sentence rather than a keyword in my opinion.

3.       Line 35-36: can you provide more detailed information about wi-fi policies on trains and buses (when and where they have been adopted, with some references)?

4.       Some additional analysis about residential location choices could be provided: a dedicated table/graph to show their attitudes – they are using more or less public transport only for lack of alternatives? How’s the attitude toward public transport of those living in more densely populated areas? Is it connected with their actual usage?

5.       Check the instructions for authors for the references embedded in the text. Remove the comma after the first author when citing him/her (for instance, ” Ben-Elia, et al.” or ” Lyons, et al.” and so on).

Author Response

We would like to thank you for what was obviously a very careful reading of our paper, and for the comments which you have provided. Please see the below responses for your comments.

Comment 1. “Probably shorten the title – you can add more keywords instead. For instance, the title can be something like ”Can access to the Internet on public transport encourage more commuters to use it regardless of their attitude?” or something similar. Think about it.”

Authors: Thank for your constructive comment. We shortened our title to “Can accessing the Internet while travelling encourage commuters to use public transport regardless of their attitude?”

Comment 2: “The keyword ”internet use while commuting” could be replaced with something else – it is more a sentence rather than a keyword in my opinion.”

Authors: Corrected.

Comment 3. “Line 35-36: can you provide more detailed information about wi-fi policies on trains and buses (when and where they have been adopted, with some references)?”

Authors: We included an example with a reference (see page 1). There are several cases for buses but we couldn’t find academic papers. For example, The First bus in the U.K. provides free Wi-Fi on their bus (https://www.firstgroup.com/tech-bus/free-wi-fi).

Comment 4. “Some additional analysis about residential location choices could be provided: a dedicated table/graph to show their attitudes – they are using more or less public transport only for lack of alternatives? How’s the attitude toward public transport of those living in more densely populated areas? Is it connected with their actual usage?”

Authors: Thanks for the very constructive comment. We agree that residential location choices are related to attitudes and travel behaviour. There are a large number of papers, investigating residential self-selection and the joint decision of residential location and travel behaviour. However, our focus here is the connection between Internet use on the move and commuting mode choice. We also considered residential locations (i.e., large urban areas, other urban areas, town and rural areas) in our model to control for their effects on commuting mode choice. We feel it is not easy to add an addition analysis about residential location choices without loosing our focus, which could result in confusions for readers.  In addition, Table 1 shows a broad picture about the relationship between attitudes and residential locations. For example, about 60% of Likers live in large urban areas (most dense areas) compared to 55% of Dis-Likers do. This implies that people who prefer public transport to private cars are more likely to live in dense areas. We indicated it in the manuscript (see page 5).

Comment 5. “Check the instructions for authors for the references embedded in the text. Remove the comma after the first author when citing him/her (for instance, ” Ben-Elia, et al.” or ” Lyons, et al.” and so on)”.

 Authors: Thanks for the comment. We downloaded Sustainability style for Endnote from the website and use it for our paper.

Round  2

Reviewer 1 Report

The issue proposed and discussed in manuscript is orginal and interesting with reference to urban transport in contect to Interenet use. In my opinion the study should be accepted for publication in its present version.